# Algorithms with Area under the Curve for Daily Urinary Estrone-3-Glucuronide and Pregnanediol-3-Glucuronide to Signal the Transition to the Luteal Phase

**DOI:** 10.3390/medicina58010119

**Published:** 2022-01-13

**Authors:** Stephen J. Usala, María Elena Alliende, A. Alexandre Trindade

**Affiliations:** 1Department of Internal Medicine, Texas Tech University Health Sciences Center, 1400 S. Coulter Street, Amarillo, TX 79106, USA; 2Programa de Cuidado y Estudio de la Fertilidad (PROCEF), Departamento de Obstetricia, Ginecología y Biología de la Reproducción, Universidad de los Andes, Monseñor Alvaro del Portillo 12455, Santiago 7620001, Chile; melenalliende@gmail.com; 3Department of Mathematics and Statistics, Texas Tech University, 1108 Memorial Circle, Lubbock, TX 79409, USA; alex.trindade@ttu.edu

**Keywords:** estrone-3-glucuronide, E3G, pregnanediol-3-glucuronide, PDG, fertility assessment methods, fertility awareness methods, FAMs, natural family planning, NFP

## Abstract

*Background and Objectives*: Home fertility assessment methods (FAMs) for natural family planning (NFP) have technically evolved with the objective metrics of urinary luteinizing hormone (LH), estrone-3-glucuronide (E3G) and pregnanediol-3-glucuronide (PDG). Practical and reliable algorithms for timing the phase of cycle based upon E3G and PDG levels are mostly unpublished and still lacking. *Materials and Methods*: A novel formulation to signal the transition to the luteal phase was discovered, tested, and developed with a data set of daily E3G and PDG levels from 25 women, 78 cycles, indexed to putative ovulation (day after the urinary LH surge), Day 0. The algorithm is based upon a daily relative progressive change in the ratio, E3G-AUC/PDG-AUC, where E3G-AUC and PDG-AUC are the area under the curve for E3G and PDG, respectively. To improve accuracy the algorithm incorporated a three-fold cycle-specific increase of PDG. *Results*: An extended negative change in E3G-AUC/PDG-AUC of at least nine consecutive days provided a strong signal for timing the luteal phase. The algorithm correctly identified the luteal transition interval in 78/78 cycles and predicted the start day of the safe period as: Day + 2 in 10/78 cycles, Day + 3 in 21/78 cycles, Day + 4 in 28/78 cycles, Day + 5 in 15/78 cycles, and Day + 6 in 4/78 cycles. The mean number of safe luteal days with this algorithm was 10.3 ± 1.3 (SD). *Conclusions*: An algorithm based upon the ratio of the area under the curve for daily E3G and PDG levels along with a relative PDG increase offers another approach to time the phase of cycle. This may have applications for NFP/FAMs and clinical evaluation of ovarian function.

## 1. Introduction

Urinary E3G (estrone-3-glucuronide) and PDG (pregnanediol-3-glucuronide) levels to assess the time of cycle are now available for fertility assessment methods (FAMs) and natural family planning (NFP) [1,2,3,4,5,6,7,8,9,10]. The devices for home measurement of E3G [4,5,8,9,10] until recently provided only qualitative results to indicate fertility. Two newly marketed devices, the Mira^TM^ and Inito^TM^, now show the actual E3G, PDG, and luteinizing hormone (LH) levels [9,10]. The algorithms to identify the fertile phase using E3G, PDG, and LH are proprietary and remain unpublished, although it is probable that thresholds, magnitudes of increase, and hormonal averages over several cycles for a specific patient are utilized in the devices. In general, these home meters are not programmed to avoid pregnancy for FAMs/NFP, and with this application they have a relatively high failure rate [4,5].

Urinary PDG test strips are now available with thresholds of 5 μg/mL and 7 μg/mL [6,7]. The concept of PDG threshold to signal the luteal phase and the safe days for FAMs/NFP is straight-forward, but with the 5 μg/mL and 7 μg/mL strips, Bouchard et al. [6] reported only 82.4% and 59% of the cycles had a positive PDG test, respectively. When using at least one positive PDG result following a urine LH positive or peak mucus sign, the average number of safe luteal days was 8.8 days [7]. The combined use of E3G and PDG with a home monitor to avoid pregnancy by NFP/FAMs is still in development.

We hypothesized that an algorithm using both E3G and PDG levels—a ratio which avoids variability due to urinary concentrations at home/point-of-care testing—could be more sensitive and specific in signaling time of cycle. We initially aimed for an algorithm that did not depend on a threshold, but rather a rate of change function. Furthermore, available for this development was a large data set of day-specific E3G and PDG levels from 25 women, 78 cycles. We found a computation using as variables the area under the curve for E3G and PDG, E3G-AUC and PDG-AUC, respectively, as the cycle progresses reliably signaled the transition to the luteal phase. Furthermore, we show this algorithm could be used as an indicator for the start of the luteal safe days for NFP.

## 2. Materials and Methods

### 2.1. Urinary LH, E3G, and PDG Day-Specific Levels

Day-specific levels of urinary LH, E3G, and PDG throughout 78 ovulatory cycles from 25 women were from a secondary analysis of data from a multicentre World Health Organization (WHO) sponsored study (HRP#87904, approved locally, 21 September 1998). Alliende et al. [11], used in-house non-competitive radioimmunoassays (reagents and assay protocols supplied by Matched Reagents Programme of the WHO) to measure urinary LH, E3G, and PDG. Day 0, the day of putative ovulation, was indexed to the day after the urinary LH rise adjusted for excretion rate (i.e., urine time interval and volume). The E3G and PDG data used herein were not adjusted for excretion rate.

### 2.2. Area under the Curve Computations for Day-Specific E3G and PDG Levels

For any given day, D, of the cycle, the area under the curves for E3G and PDG were computed using Microsoft Excel or GraphPad Prism version 9.2 for Windows, GraphPad software, San Diego, CA, USA (www.graphpad.com accessed on 2 January 2022), which give identical results. E3G-AUC, PDG-AUC, and the corresponding Delta values (see below) are dependent on the initial day (i.e., start day) for the computation.

### 2.3. Computations of the Delta Values: Delta5, Delta6, Delta7

The ratio, E3G-AUC/PDG-AUC, is computed for day, D-1, of the cycle, and similarly calculated for the subsequent day, D. The Delta value for day, D, is then the difference:Delta on day, D = (E3G-AUC/PDG-AUC on Day, D) − (E3G-AUC/PDG-AUC on Day, D-1)

For Delta value computations, Delta5 (‘D5’), AUC calculations begin on Day 6 of the cycle, for Delta6 (‘D6’) on Day 7, and for Delta7 (‘D7’) on Day 8. That is, urinary E3G and PDG levels are used starting on calendar Day 6, Day 7, and Day 8 for D5, D6, and D7, respectively. The Delta5 nomenclature was assigned since the first five days of urinary hormone levels were not used in the D5 computation, and in a similar fashion for D6 and D7. Therefore, the first D5 value was recorded on calendar Day 7 since it is a difference of the E3G-AUC/PDG-AUC ratio between Day D and Day D-1, and similarly the first D6 and D7 values were recorded on calendar Days 8 and 9, respectively. In this manner, day-specific D5, D6, and D7 values were generated.

### 2.4. The D5D6D7 Convolution

D5, D6, and D7 values were determined for all the cycles. The signs of these Delta values were considered together—a convolution called D5D6D7—which served as an improved indicator for time of cycle. The D5D6D7 value is a sign determination: a sign of positive (denoted as ‘0’) is assigned on a cycle day, if any of the D5, D6, D7 values are positive and a sign of negative (denoted as ‘1’) only if D5, D6, and D7 values are all negative. Consequently, for every day of the cycle D5D6D7 is mapped to a ‘0’ or ‘1’ starting on calendar Day 9, which was the first day of combined D5, D6, and D7 values.

### 2.5. The D5D6D7 Convolution with PDG Modifier to Signal the Start of the Luteal Phase Safe Days

To improve specificity in identification of the luteal transition and practicality of the algorithm for marking the start of the luteal safe days, a cycle-specific PDG increase from baseline was derived and added to the D5D6D7 analysis. The PDG modifier was only applied in a sequence of negative D5D6D7 values (‘1’s) of three or greater length. The PDG modifier—called the ‘5dP-3x’ rule—is applied starting on the third day of any sequence of ‘1’s for D5D6D7 values. For any negative D5D6D7 sequence starting on Day D, and progressing D + 1, D + 2, etc., a cycle-specific baseline is calculated as the mean PDG for the five days ending two days before the start of the negative sequence. In other words: if Day D, is the start of the negative sequence, then the mean PDG is computed for days D-2 to D-6, inclusive. A positive PDG modifier signal occurs if a PDG level is 3-fold or above this baseline on any days D, D + 1, D + 2, etc., of the negative sequence. This interrogation begins with the 3rd day of any negative D5D6D7 sequence (sequence of ‘1’s) and continues as the cycle progresses in real-time until two positive PDG modifier signals occur. The start of the safe luteal days is set for the day of the 2nd positive PDG modifier signal. The PDG modifier interrogation of a negative sequence continues until the sequence turns positive (a ‘1’ to a ‘0’) or until the 2nd positive PDG modifier.

### 2.6. Statistical Evaluation of the Algorithms

In the *p*-value calculations that assess the probability of a particular event (say, Event A) occurring by chance alone (the null hypothesis), our reference model for the null in any given cycle is a random binary sequence with equal probabilities of success or failure (a coin toss, or equally weighted independent Bernoulli trials). The window of opportunity for Event A to happen is always taken to be the time span of Day −2 to Day +10 (the common time period for all cycles). If Event A happens in n out of N cycles, then the overall probability constituting the final *p*-value, is simply modeled as a binomial random variable [12].

## 3. Results

### 3.1. The Delta Function Signals the Transition to the Luteal Phase by an Indicator Negative Sequence

A novel method was discovered that marked the periovulatory transition interval. This was developed from day-specific urinary E3G and PDG levels from 25 women, 78 cycles with a range of cycle lengths, 23 to 35 days (mean 28.5 ± 2.5 (SD)), and a mean calendar day for ovulation of 15.5 ± 2.4 (SD). The E3G and PDG data, as E3G-AUC/PDG-AUC ratios, were used to compute the day-specific D5, D6, D7 values. In all 78 cycles, an extended, uninterrupted sequence of negative D5, D6, and D7 values—nine or greater in length—began in the ovulatory to luteal transition interval (Appendix A). This extended negative sequence of Delta values, called the indicator negative sequence (INS) identified the luteal phase in all 78 cycles. An example of the INS for D5, D6, and D7, is shown in Table 1 for cycle 20.1.

### 3.2. Convolution of Delta5, Delta6, and Delta7- ‘D5D6D7’—As an Improved Indicator of the Transition to the Luteal Phase

The D5, D6, and D7 values always underwent an extended sequence of nine negative values or more to mark the luteal phase. However, negative sequences of lesser length occurred in the preovulatory phase in some of the cycles which would confound early identification of the start of the safe days (see Appendix A and Table 1). Therefore, a further methodology was sought with the Delta function to reduce the occurrence of these lesser negative sequences.

It is known that there can be preovulatory LH surges and unsustained estrogen rises [13,14,15,16,17]. We speculated these hormonal variations cause some of the early preovulatory negative D5, D6, and D7 values. To reduce the signal from non-ovulatory fluctuations, it was hypothesized that the D5, D6, and D7 computations could be combined—a convolution called D5D6D7—generating a positive or negative sign which was given a ‘0’ or ‘1’ value, respectively. Consequently, given that the ratio E3G/PDG should generally increase with follicular development until the periovulatory interval, a true negative D5D6D7 was scored only if D5, D6 and D7 were all negative (‘1’). If either D5 or D6 or D7 were positive—that is, not all of negative sign—then D5D6D7 was scored as positive (‘0’). In this way, it was thought that the signal from early unsustained follicular development in an extended cycle mimicking high then low estrogen levels would be suppressed by absence of complete D5, D6, and D7 negative signs. In summary, the sign assignment by the D5D6D7 convolution analysis—the mapping of D5, D6, and D7 to either a ‘0’ or ‘1’—was dedicated to signaling the transition to the luteal phase. The extended sequence of negative D5D6D7 values, as a sequence of nine ‘1’s or more, was also called the INS.

With D5D6D7 convolution analysis, E3G and PDG urine samples begin on calendar Day 6, but D5D6D7 signs are not considered for scoring until calendar Day 9, when D5, D6 and D7 values are all available. To better see the assignment of the D5D6D7 value (‘1’ if D5, D6, D7 all negative, ‘0’ if any D5, D6, D7 positive), an example of this convolution analysis for cycle 20.1 is provided in Table 1.

The D5D6D7 values for all 78 cycles are provided in the Appendix A.

D5D6D7 analysis did filter out many of the negative values in the preovulatory phase, but some remained that were up to 6 days in length (5/78 cycles). A single INS, a sequence of ‘1’s which was 9 or longer in length starting in the periovulatory interval and continuing into the luteal phase, was maintained by D5D6D7 analysis; an INS of 9–17 length was found in 78/78 cycles. The first day of the INS was: Day −1, 2 cycles (2.6%); Day 0, 22 cycles (28.2%), Day +1, 39 cycles (50%), Day +2, 13 cycles (16.6%), Day +3, 2 cycles (2.6%) (Figure 1). Using as reference background for the null hypothesis a binary sequence of ‘0’ or ‘1’ having equal probabilities (0.5), the probability of a true INS (a consecutive sequence of all ‘1’ of length ≥9) starting on Day −1 by chance in 78/78 cycles is <0.01.

### 3.3. A PDG Modifier with D5D6D7 to Optimize Identification of the Start of the Safe Days

However, one important aim of the algorithm was not only to identify by INS the luteal phase, but also to establish the start of the safe luteal period for FAMs/NFP as soon as possible, not requiring a nine day wait. To accomplish this aim, an additional step was incorporated into the D5D6D7 convolution analysis called the ‘5dP-3xrule’ (five-day baseline, at least three-fold PDG increase). This algorithmic step included a start of the safe luteal period signal of at least a three-fold increase in PDG from a cycle-specific baseline during days of extended sequences (three or greater) of negative D5D6D7.

The cycle-specific five-day PDG baseline levels preceding negative sequences, the INS and any preovulatory sequences three or greater in length, with the corresponding PDG levels for the start of the luteal safe period are provided in the Appendix A. The mean five-day PDG baseline for the INS sequences was 1.2 μmol/L ± 0.6 (SD), range 0.2–2.6 μmol/L, and the mean safe day start PDG level was 7.6 μmol/L ± 5.0 (SD), range 1.9–31.4 μmol/L. The start of the luteal safe days for the 78 cycles are shown in Appendix A and for cycle 20.1 in Figure 1. The 5dP-3x rule correctly identified a post-ovulatory, ‘safe start day’ in 78/78 cycles:Day + 2, 10 cycles (12.8%); Day + 3, 21 cycles (26.9%); Day + 4, 28 cycles (35.9%).Day + 5, 15 cycles (19.2%); Day + 6, 4 cycles (5.1%). Using as reference background for the null hypothesis a random binary sequence with equal probabilities (0.5), the probability of a true post-ovulatory day occurring on Day + 2 or later by chance in 78/78 cycles is <0.01.

For these 25 women, 78 cycles: the mean INS start day was Day 0.9 ± 0.8 (SD); mean INS length in days, 12.6 ± 1.5 (SD); mean start day for the luteal safe period, Day 3.8 ± 1.1 (SD); and mean number of safe luteal days, 10.3 ± 1.3 (Figure 1).

### 3.4. Signature of Delta Function Plots

The D5D6D7 convolution is not an averaging of the magnitudes, however a plot of the mean day-specific D5, D6 and D7 values starting from calendar Day 9 is informative and presented in Figure 2, showing a characteristic signature for the Delta function.

## 4. Discussion

The rationale of this work was to develop an algorithm with the new technology of urinary E3G and PDG measurements for identifying time of cycle for NFP purposes. An algorithm refers to a step-by-step method for performing some action [18]. For an algorithm one frequently uses a sequence of instructions repeatedly, until an objective is reached. As simply put by Lewis Carroll in *Alice’s Adventures in Wonderland*: ‘Begin at the beginning…and go on till you come to the end: then stop’ [19].

Given biological fluctuations, the development of a highly sensitive and specific algorithm to determine the phase of cycle for an individual woman is more difficult than for the ‘average’ cycle using mean hormonal data [20,21]. To accomplish such an algorithm, computations with urinary E3G and PDG levels from 78 cycles from 25 women using real-time AUCs as variables were created to establish methods to signal the transition from the preovulatory to the luteal phase. This algorithm utilized a D5D6D7 convolution, which generated a sequence of negative signs greater than nine in length, denoted as a sequence of ‘1s’ and called the INS, to identify the luteal phase. The D5D6D7 method does not require a threshold or percentage reduction from a peak and a E3G/PDG based algorithm avoids fluctuations in urinary concentration at home/point of care testing. However, one main objective was to create a practical method for NFP, and therefore, an increase above a three-fold threshold of a cycle-specific PDG baseline was incorporated into the D5D6D7 algorithm. It should be noted that in regularly menstruating women there is variability in the luteinization process reflected in the rate and level of the PDG rise, making it difficult to use a universal PDG threshold to identify the early post-ovulatory time [22]. One strength of this three-fold PDG modifier is that it is calculable in every cycle, independent of any past cycle data, and less likely to err since it is only employed on negative D5D6D7 days (i.e., ‘candidate’ luteal days).

A format to program D5D6D7 convolution analysis with PDG modifier into a home meter to signal the luteal transition and the start of the luteal safe days is outlined in Figure 3. With the algorithm the luteal transition interval was identified in all 78 cycles. In addition, the mean number of derived safe luteal days for NFP was 10.3 ± 1.3 (SD), and the luteal phase predicted by Day + 4 in 75.6% and by Day + 6 in 100% of cycles. Although this study was not a clinical trial and therefore cannot be directly compared with other NFP publications [5,6,7], the results indicate possible improvement in timing the start of the luteal phase and shortening the days of abstinence. The D5D6D7-PDG modifier algorithm could be incorporated into home monitors with proprietary E3G, PDG, and LH algorithms to improve signaling of the luteal phase.

The algorithm as presently construed lacks the capability to indicate the start of the preovulatory fertile period. Given this limitation, until detection of the INS by a device, a woman would need to consider the time of cycle as preovulatory. For NFP other FAMs such as cervical-vaginal mucus signs could be used to indicate preovulatory fertile days or perhaps with other methodologies yet to come.

D5D6D7 convolution analysis could potentially provide information on abnormal cycles such as in polycystic ovary syndrome. In 5 of 78 cycles (6.4% of cycles), there was a preovulatory negative D5D6D7 sequence, ‘1’s of 3–6 length. These cycles were of longer length (30–35 days) and later ovulation (calendar Days 15–23). As mentioned earlier, it is possible that these preovulatory days represented attempts of early ovulation which did not progress [13,14,15,16,17]; it would be interesting in future studies to link data of daily serum or urinary LH levels with D5D6D7 analysis. In addition, it will be important to test the D5D6D7 method with PCOS and breast-feeding cycles to determine if it can identify both an extended preovulatory state and a post-ovulatory, luteal state. Furthermore, the precise length of the INS may have implications in terms of the health of the luteal phase. It is hoped that D5D6D7 convolution with PDG modifier could be applied as a tool for both NFP/FAMs and the study of dysfunctional ovulation and inadequate luteal phase PDG levels.

## 5. Conclusions

An algorithm using a progressive negative change in the daily ratio of the area under the curve for E3G and PDG, E3G-AUC/PDG-AUC along with a three-fold increase in PDG from a cycle-specific baseline provides a sensitive and specific signal for the transition to the luteal phase in ovulatory cycles. In addition, the start of the safe period for NFP/FAMs can be identified. Since E3G and PDG levels from available home fertility monitors can be input into the algorithm, it may be of use for women interested in understanding their luteal phase and in planning abstinence for NFP.

## Figures and Tables

**Figure 1 medicina-58-00119-f001:**
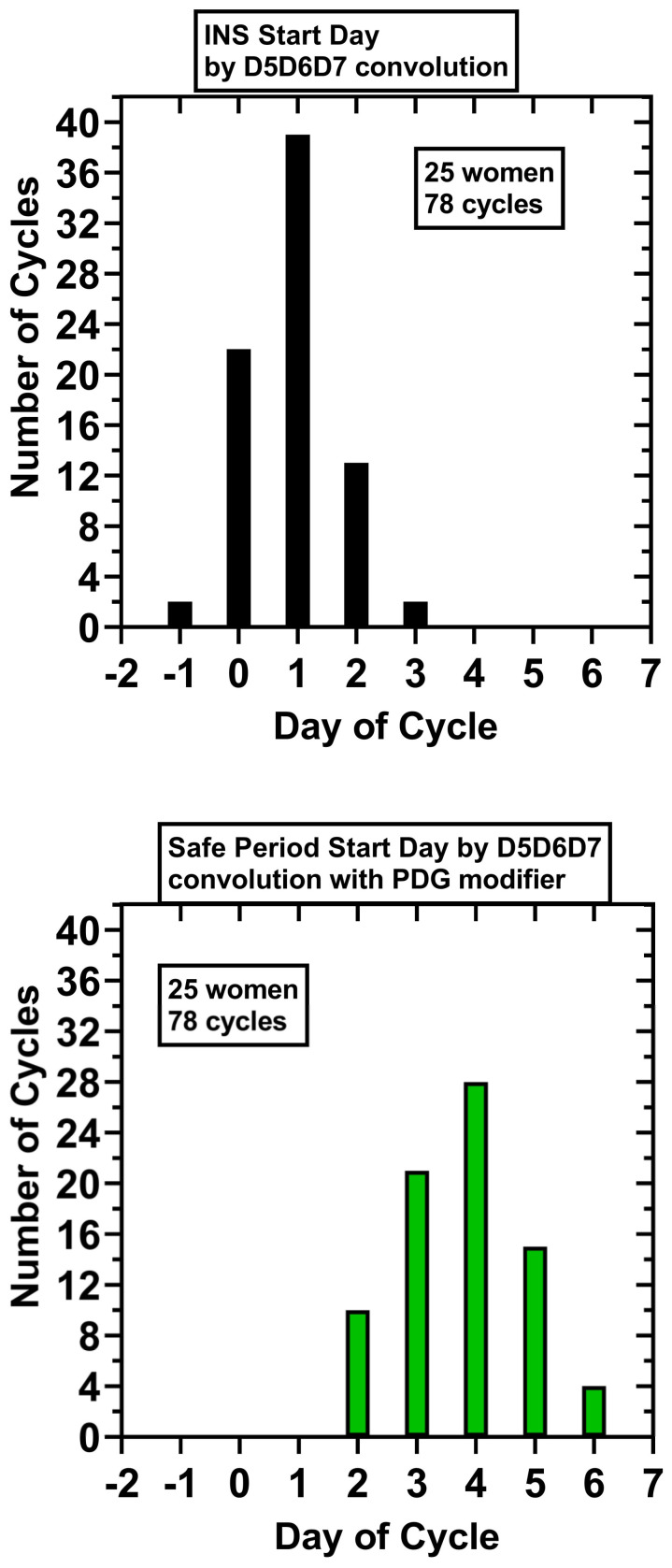
Start days of the indicator negative sequence (INS) by D5D6D7 convolution analysis and the start days for the luteal safe period by D5D6D7 convolution analysis with addition of a PDG modifier. Day of cycle is indexed to the day of putative ovulation, Day 0, the day after the urinary LH surge. A total of 78 cycles from 25 women were analyzed by a D5D6D7 convolution as described in the Section 2 and Section 3, generating a sequence of negative values 9–17 in length, the INS, which started in the periovulatory interval and extended into the luteal phase (**top**); this served as a signal for the luteal phase transition. A ≥3-fold increase in PDG from a cycle-specific baseline was used on the days of the INS to mark the start of the luteal safe days with the ‘5dP-3x’ modifier as described in the Section 2 (**bottom**).

**Figure 2 medicina-58-00119-f002:**
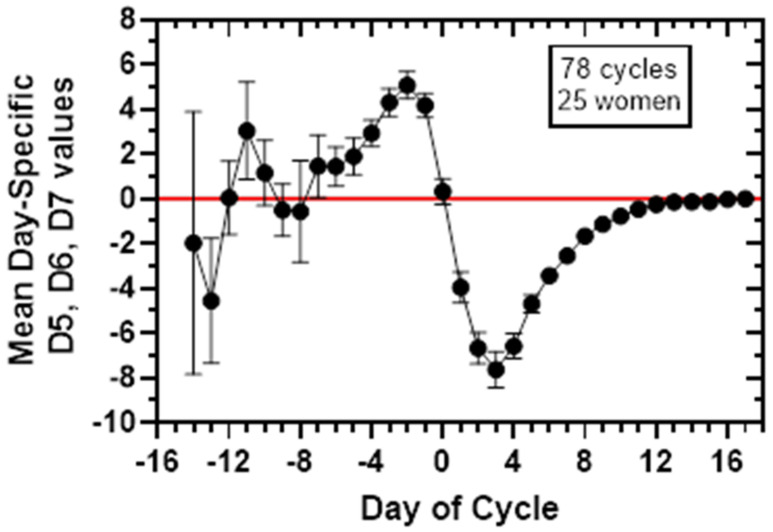
Mean day-specific Delta5, Delta6, Delta7 values taken together with 95% CIs for 78 cycles from 25 women. The day of the cycle is indexed to the day of putative ovulation, Day 0, the day after the urinary LH surge. The Delta value for a cycle on Day D, is the difference in the ratio E3G-AUC/PDG-AUC between day, D, and the preceding day, D-1, where E3G-AUC and PDG-AUC are the area under the curve for E3G and PDG, respectively, on a given day, D. Delta5 (D5), Delta6 (D6), Delta7 (D7) are the AUC computations beginning on calendar Day 6, Day 7 and Day 8 of a cycle, respectively. The mean day-specific values shown here used the D7, D6 and D5 values starting at calendar Day 9. These Delta values were used for the D5D6D7 convolution as described in the Section 2 and are provided in Appendix A.

**Figure 3 medicina-58-00119-f003:**
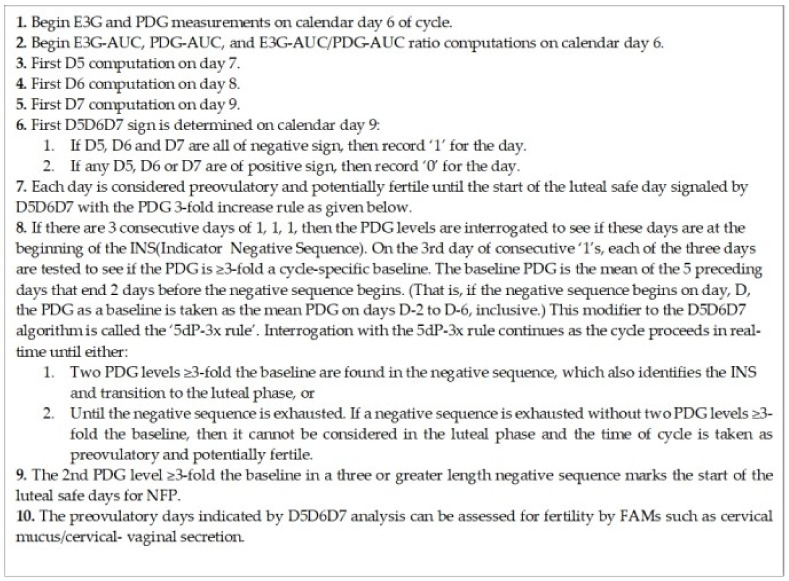
Summary of the algorithm for the D5D6D7 convolution analysis with the PDG modifier. Steps are given for computer programming of a home E3G, PDG monitor which signals the transition to the luteal phase and the start of the luteal safe days for NFP. The D5, D6, and D7 (Delta computations) are described in the Section 2 and Section 3.

**Table 1 medicina-58-00119-t001:** D5D6D7 convolution analysis with application of PDG modifier for cycle 20.1 *.

Day of Cycle	D7	D6	D5	D5D6D7	Start of the Luteal Safe Days by D5D6D7 with PDG Modifier
−12	−1.56	−0.63	0.35	0	0
−11	6.10	4.55	3.62	0	0
−10	10.21	7.98	6.42	0	0
−9	1.98	2.26	2.45	0	0
−8	−0.50	0.27	0.90	0	0
−7	−2.61	−1.64	−0.71	1	0
−6	−3.08	−2.35	−1.58	1	0
−5	−1.16	−0.84	−0.44	1	0
−4	0.88	0.94	1.05	0	0
−3	2.89	2.71	2.55	0	0
−2	7.28	6.76	6.25	0	0
−1	10.25	9.59	8.92	0	0
0	4.65	4.47	4.30	0	0
1	**−2.52**	**−2.02**	**−1.47**	**1**	0
2	**−2.57**	**−2.18**	**−1.72**	**1**	0
3	**−4.06**	**−3.67**	**−3.19**	**1**	**1**
4	**−10.95**	**−10.18**	**−9.23**	**1**	0
5	**−9.54**	**−9.07**	**−8.45**	**1**	0
6	**−6.01**	**−5.81**	**−5.52**	**1**	0
7	**−4.51**	**−4.40**	**−4.24**	**1**	0
8	**−2.83**	**−2.78**	**−2.70**	**1**	0
9	**−1.46**	**−1.45**	**−1.41**	**1**	0
10	**−0.90**	**−0.89**	**−0.87**	**1**	0
11	**−0.60**	**−0.60**	**−0.59**	**1**	0
12	**−0.35**	**−0.35**	**−0.34**	**1**	0

* Day 0 (shaded grey), the day of putative ovulation, is the day after the urinary LH surge. The D7 value for Day D, is the difference in the ratio of E3G-AUC/PDG-AUC between Day D, and the preceding day, D-1. E3G-AUC and PDG-AUC are the areas under the curve for E3G and PDG, respectively, on a given day, D. The D7 calculations were all performed starting with E3G and PDG levels on calendar Day 8. The D6 and D5 computations were similarly performed except starting with E3G and PDG levels on calendar Days 7 and 6, respectively. The D5D6D7 convolution is a sign assignment on a cycle day as ‘0’ if any D5, D6, D7 are positive on that day and ‘1’ only if all D5, D6, D7 are negative. The D5D6D7 sign assignment begins on calendar Day 9. The indicator negative sequence (INS), as extended negative D5, D6, and D7 values and as a sequence of ‘1’s nine in length or greater, which occurs in the luteal phase, is shown in bold. The start of the safe days was by the ‘5dP-3xrule’ as described in the text. E3G, estrone-3-glucuronide; PDG, pregnanediol-3-glucuronide; LH, luteinizing hormone.

## Data Availability

Data available in the Appendix A.

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
