# Peer review of "Algorithms with Area under the Curve for Daily Urinary Estrone-3-Glucuronide and Pregnanediol-3-Glucuronide to Signal the Transition to the Luteal Phase"

_medicina, 2022, doi:10.3390/medicina58010119_

Round 1

Reviewer 1 Report

1) The abstract has too many technical details about the algorithm calculation. The population could be briefly described, and the algorithm could be more simply summarized.

2) The introduction needs to be updated regarding current monitors that measure E3G & PDG (p. 2, line 60), Mira, Inito and Oova all have PDG quantitative measurements.  Mira and Into have E3G measurements, all 3 have LH measurements.

3) Section 2.3 needs to be re-written.  There is some lack of clarity between day 1 of the cycle and Day 0 relative to ovulation.  The Delta days could be more clearly defined without having to give an excess of examples for multiple days.  It may even be better explained in a simple table.

4) Section 2.4 also requires more clarity.  I would suggest defining a new term rather than D5D6D7 which makes reading challenging.  Moreover the references to "sign of positive" and "sign of negative" are not clear and need to be better explained.  An explanatory table could help both 2.3 and 2.4 be clearer for the reader.

5) There is an underlying assumption that needs to be challenged by the data - in section 3.2, they comment that the E3G/PDG ratio should generally increase with follicular development, this may not be actually true.  Since this assumption underpins D5D6D7 and the creation of a positive or negative sign, the authors should reconsider their classification scheme.  The E3G values shown with various quantitative monitors show fluctuations in E3G in the follicular phase that do not fit a typical linear pattern.

6) I find the term "convolution analysis" an unnecessary neologism and complicates the explanation of the statistics.

7) I would have preferred seeing a cumulative AUC approach with each day added to the sum of the previous days, rather than the diad of Delta D and D-1.   

8) I would have liked to see an attempt to analyze the beginning of the fertile window with a cumulative AUC approach.  Rather than only define the end of the fertile window, the AUC may have also been proposed as a way to identify the start of the fertile window.

9) The effectiveness of NFP methods is highly dependent on defining the post-ovulatory infertile phase.  The proposition of identifying an "early-post ovulatory time" 271-272 should be really referenced to ultrasound (at least comment on this in the Discussion) to ensure it is conservative enough for NFP users.

10) The Discussion, like the Intro, needs to be updated to identify that there are 3 quantitative monitors that are available to be used as described above.

Reviewer 2 Report

Thank you very much for this interesting manuscript. It is an important step on the improvement of fertility/infertility management road.

The title, introduction and the methods sections are well-written. The study results are interesting and well-described in the RESULTS section. 

However, the Discussion part could be improved a lot.

The researchers performed a very interesting study. Please discuss it in this section appropriately.  Please restructure this part to cover the following:

1. Rationale of the study (why it was done)                                      

2. Main findings of the study

3. What makes your study unique

4. What it adds to what we already know

5. Comparison of your results with previously published ones. Agreement and disagreement with the studies compared.

6. Please define clinical implication of your study findings and discuss it in the Discussion section.

7. Strength and limitations

The conclusion part should be more elaborative. Please expand it.

Round 2

Reviewer 1 Report

-Recommend removing reference to "safe luteal days" and specific +2, +3, +4 etc days in abstract because this is not an effectiveness study.

-Change "a home reality" line 49 to "available"

-Change line 52 "a 'yes' or 'no'" to "qualitative results"

-Section 2.5 is a post-hoc analysis that was explained in the results as well.  The methods and results should specify that this is a post-hoc analysis that was identified in the process of doing the study

-Line 266 - I still don't find the Lewis Carroll quote helpful

-Any reference to safe days throughout the paper is inappropriate because safe days imply effectiveness at avoiding pregnancy.  I don't understand how it would be possible to indicate different safe days (+2, +3, +4, +5, etc) based simply on an algorithm that has not been tested on women avoiding or achieving pregnancy.

Author Response

-Recommend removing reference to "safe luteal days" and specific +2, +3, +4 etc days in abstract because this is not an effectiveness study.

We understand the reviewer’s concern here and have changed the abstract to ‘predicted the start of the safe period…’ This is version 3 (v3).

Yes, the algorithm was derived from the data – that is, fit to the data, and this is clear in the manuscript. The strength is that the derivation is not based on just 10 cycles but rather 78 cycles from 25 women. But that is how all the ‘rules’ for NFP began, such as the ‘3-day rule’ for BBT, etc. We had already made this clear in the revised discussion (v2 and v3) that this is not a clinical trial and therefore cannot be compared with other NFP studies (3rd paragraph discussion). We hope that this algorithm or some variant thereof can be applied in future clinical studies.

(Please make sure you are reading the correct abstract – the one directly on the uploaded m.s. v3.)

-Change "a home reality" line 49 to "available"

Done.

-Change line 52 "a 'yes' or 'no'" to "qualitative results"

Done and the sentence was changed.

-Section 2.5 is a post-hoc analysis that was explained in the results as well.  The methods and results should specify that this is a post-hoc analysis that was identified in the process of doing the study

We have added the word ‘derived and added…’ to line 105 to make this point.

-Line 266 - I still don't find the Lewis Carroll quote helpful

This is a matter of taste. This quote was actually seen in the reference on algorithms. It emphasizes what an algorithm is to readers unfamiliar with the concept.

-Any reference to safe days throughout the paper is inappropriate because safe days imply effectiveness at avoiding pregnancy.  I don't understand how it would be possible to indicate different safe days (+2, +3, +4, +5, etc) based simply on an algorithm that has not been tested on women avoiding or achieving pregnancy.

This is discussed above, and to further make this point we have changed the Discussion to:

(Line 269):  In addition, the mean number of derived safe luteal days for NFP was 10.3±1.3(SD), and the luteal phase predicted by Day +4 in 75.6% and by Day +6 in 100% of cycles. Clearly, the algorithm would need to be tested in a clinical trial.